# Moral foundations theory, political identity, and the depiction of morality in children's movies

**Rachel Gehman**[1], **Steve Guglielmo**[2]*, **David C. Schwebel**[3]

**1** Department of Psychological Science, University of Vermont, Burlington, VT, United States of America,
**2** Department of Psychology, Macalester College, St. Paul, MN, United States of America, **3** Department of
Psychology, University of Alabama at Birmingham, Birmingham, AL, United States of America

* sgugliel@macalester.edu

UNITED KINGDOM

**Data Availability Statement:** All data, materials,
and code are available from the Open Science
Network: http://doi.org/10.17605/OSF.IO/2TGM8.

**Funding:** The authors received no specific funding
for this work.

## Abstract

Children's movies often provide messages about morally appropriate and inappropriate con-
duct. In two studies, we draw on Moral Foundations Theory (MFT) to derive predictions
about actual depictions of morality, and people's preferences for different moral depictions,
within children's movies. According to MFT, people's moral concerns include individualizing
foundations of care and fairness and binding foundations of loyalty, authority, and sanctity.
Prior work reveals that although there are political differences in the endorsement of these
two broad categories—whereby stronger political conservatism predicts stronger binding
concerns and weaker individualizing concerns—there nonetheless is broad agreement
across political identity in the importance of individualizing concerns. We therefore predicted
that heroes would value individualizing foundations more than villains, and that despite polit-
ical differences in preferences for moral messages, there would be more agreement in the
importance of messages promoting individualizing concerns. In Study 1, we coded heroes
and villains from popular children's movies for their valuation of moral foundations. Heroes
valued individualizing concerns more, and binding concerns less, than villains did. Partici-
pants in Study 2 considered moral dilemmas faced by children's movie characters, and
rated their preferences for resolutions that promoted either individualizing or binding founda-
tions. Although liberals preferred individualizing-promoting resolutions and conservatives
preferred binding-promoting resolutions, there was stronger agreement across political
identity in the importance of individualizing concerns. Despite political differences in moral
preferences, popular depictions of children's movie characters and people's self-reported
moral endorsement suggest a shared belief in the importance of the individualizing moral vir-
tues of care and fairness.

Movies are often infused with moral messages. From their exploration of overarching
themes, their ascription of particular traits to heroic and villainous characters, and their reso-
lution of pivotal moral dilemmas, movies provide viewers with depictions of morally virtuous
(and morally suspect) behavior. Moral messaging in children's movies is of particular impor-
tance, since it is targeted at an audience for which morality is actively developing. What
moral messages do filmmakers (and consumers, including parents) want children's movies
to depict? Are these preferences related to people's political identity? And what are the

**Competing interests:** The authors have declared
that no competing interests exist.

actual moral depictions presented in movies? In the present two studies, we draw on an
influential theory of moral judgment—Moral Foundations Theory—to develop and test pre-
dictions about the depiction of morality in children's movies.

## Moral foundations and political identity

Evolutionary pressures shape organisms in part to act selfishly but also to trust and cooperate
with members of social groups [1]. Morality helps enable this cooperation, as shared moral
concerns create expectations and accountability regarding social conduct. Moral Foundations
Theory proposes that to promote different social goals, humans possess five categories of
moral concern: care/harm, fairness/reciprocity, loyalty/ingroup, authority/respect, and sanc-
tity/purity [2].

Care and fairness constitute the superordinate category of *individualizing foundations*, pro-
moting the rights of individuals and protecting them from unfair treatment [3]. Concerns
about care/harm result from evolutionary pressures regarding protection of community mem-
bers and aversion to harm [4, 5], resulting in virtues that condemn cruelty and promote the
protection of individuals. Concerns about fairness stem from evolutionary pressures that pro-
mote cooperation and discourage free-riding, which underlie the moral emotions of guilt and
moralistic aggression that combat unfair behavior [6].

The *binding foundations* of loyalty, authority, and sanctity enable group solidarity by unit-
ing individuals to their social groups. Loyalty manifests in virtues that promote conformation
to group norms, patriotism, and self-sacrifice for the group [3]. Authority emphasizes respect
of and obedience to authority figures and hierarchical structures, such as parents or superiors.
These foundations correspond with evolutionary pressures to form and sustain stable hierar-
chical structures and cohesive group identity [7, 8]. The sanctity foundation entails sacred val-
ues that promote chastity or cleanliness [9], which are often grounded in feelings of disgust,
serving to protect humans from parasites and pathogens [10, 11] and discouraging lust, greed
and other selfish impulses [9].

Although the five moral foundations reflect shared adaptive mechanisms of moral regula-
tion, they nonetheless exhibit substantial variability, including across political identity. The
ideological left often frames moral arguments around the protection and promotion of the
rights of individuals [12], reflecting the belief that people have a right to pursue their own indi-
vidual goals [13]. The ideological right often holds that cooperative societies depend on struc-
tures that include hierarchy, institutions, and traditions to enable peace and productivity [14],
reflecting the belief that humans need structures to incentivize cooperation [13].

Moral Foundations Theory generally holds that liberals view the individualizing founda-
tions as more relevant to their understanding of morality as compared to conservatives,
whereas conservatives view the binding foundations as more important than liberals do [14].
Substantial evidence supports these patterns. For example, Graham et al. [14] asked partici-
pants to indicate which particular moral concerns were most relevant to their sense of right
and wrong. Stronger political conservatism was associated with stronger reported relevance of
the binding foundations and with weaker reported relevance of the individualizing founda-
tions. Similar patterns emerged in people's reported (un)willingness to violate particular moral
foundations for money: stronger political conservatism predicted decreased willingness to vio-
late binding foundations and increased willingness to violate individualizing foundations. Sub-
sequent studies have replicated the association between political identity and moral

foundation preferences across large, geographically diverse samples [15]. Thus, the link between political identity and moral foundation preferences is generalizable across cultural contexts, rather than specific to any particular nation or culture.

Some evidence nonetheless calls into question the consistency and scope of the evidence for MFT. Whereas Graham et al. [14] found that religious sermons from the ideological left more often emphasized individualizing foundations and sermons from the ideological right more often emphasized binding foundations, a replication effort largely failed to replicate these associations among a more diverse sample of religious denominations [16]. Further, the link between political identity and endorsement of particular moral foundations can vary across contexts. For example, in contexts that involve environmental concerns, liberals become more likely to value sanctity (of nature/Earth; [17]) and more likely to support obeying authority figures that value the environment [18].

On the whole, the preponderance of existing evidence supports the claims of MFT, which can be distilled into the following (here dubbed *Claim 1*): stronger political conservatism is associated with weaker endorsement of the individualizing foundations and stronger endorsement of the binding foundations.

A separate set of findings reveals that despite political differences in endorsement of individualizing and binding foundations, the individualizing concerns might be uniquely important across the political spectrum, at least among American participants. Graham et al. [14] showed that even among conservatives, and especially among liberals, individualizing foundations were endorsed at least as strongly as binding foundations. Wright and Baril [19] found that when participants faced cognitive load while making moral judgments—thus relying more on intuitive preferences—both liberals and conservatives viewed individualizing foundations as far more important than binding foundations. Schein and Gray [20] showed that for liberals and conservatives alike, perceptions of harm were most accessible and most predictive of moral condemnation (as compared to perceptions of other moral foundations). Frimer [16] reported that liberals and conservatives did not differ in their use of harm- and fairness-related language (whereas they differed in their use of binding-related language). Thus, even though political identity is associated with differential endorsement of individualizing and binding foundations, individualizing concerns nonetheless are viewed as especially important across political identity.

The above findings therefore support a second claim (here dubbed *Claim 2*): American participants endorse the individualizing foundations more widely than the binding foundations. An implication of this claim is that political identity is differentially associated with these two categories of moral foundations: political identity will be a weaker predictor of individualizing foundations (for which there is relatively more endorsement across the political spectrum) than of binding foundations.

## Stories and moral depictions

People's moral concerns are not mere preferences in their minds, but are also values that are socially communicated and transmitted. One critical context of moral value transmission is children's moral education and development. Children receive moral messages—from parents, peers, and broader society—as they learn to internalize the moral norms of their culture [21, 22]. Stories with moral messages are especially effective moral education tools, as they capitalize on children's moral intuitions, enabling children to immediately recognize transgressions such as cheating, disobedience, and harm again innocent people [2].

The moral content of stories can influence children's subsequent behavior. For example, Vandenplas-Holper et al. [23] found that hearing stories with altruistic protagonists, as

compared to those without a prosocial message, made children more likely to help needy peers. Similarly, Lee et al. [24] showed that after hearing stories in which the protagonist benefits from being honest, children were less willing to lie.

Stories also depict heroes and villains with particular characteristics or tendencies, thereby communicating information about whether such characteristics are moral or immoral. For example, children's movies often use physical qualities of beauty as shorthand for characters' moral integrity: attractive characters are more likely to be portrayed as "good guys", whereas unattractive characters are more likely to be overweight and to commit violent and immoral acts [25]. Villainous characters are also more likely to speak with foreign accents [26], thus implying that they are members of an outgroup.

Importantly, evidence reveals that people form perceptions of narrative characters' MFT foundations, and that they perceive heroes and villains in systematically different ways. Grizzard et al. [27] showed that people's perceptions of characters' morality conformed to a five-factor model corresponding to the dimension of MFT. In other words, people formed distinct perceptions about characters' valuations of care, fairness, loyalty, authority, and purity. Further, when characters were manipulated to seem heroic vs. villainous (based on their facial expressions, hair/outfit color, and attractiveness), people perceived the heroic character to be more moral—across all five MFT dimensions—as compared to the villainous character [28].

We extend the literature on MFT and perceptions of characters' morality by examining depictions of heroic and villainous characters in widely-viewed children's movies. Moreover, we examine not just actual depictions of morality in such movies (Study 1), but also people's preferences for different hypothetical moral depictions or messages, particularly as related to their own political identity (Study 2). By investigating these related and unanswered questions about depictions of moral characteristics and messages within children's media, the present studies aim to extend and conceptually replicate the key claims of MFT.

In Study 1, we examined moral representations in top-grossing children's movies by evaluating heroic and villainous characters' valuation of each moral foundation. Since, according to *Claim 2* above, Americans value the individualizing foundations more strongly than binding foundations, we hypothesized that individualizing foundations would be depicted as morally virtuous in American children's movies. That is, heroes would value individualizing foundations more strongly than villains, whereas the opposite would be true for binding foundations. In Study 2, we examined whether people's own preference for different moral messages or resolutions in children's movies would depend on their political identity. According to *Claim 1* above, stronger political conservatism is associated with weaker endorsement of individualizing concerns and stronger endorsement of binding concerns. In aiming to conceptually replicate these patterns to a new domain (preferences for moral resolutions within children's movies), we therefore hypothesized that stronger political conservatism would predict a stronger preference for binding-promoting moral resolutions, and a weaker preference for individualizing-promoting moral resolutions. Moreover, in accordance with *Claim 2*, we further hypothesized that, as compared to binding foundations, individualizing foundations would be more weakly correlated with political identity (reflecting relatively greater endorsement of individualizing foundations across the political spectrum).

## Study 1

Study 1 examined how the five moral foundations are portrayed in children's movies. We identified the most popular children's movies across a 70-year period and coded moral characteristics of heroic and villainous characters in each movie. Since perceptions of moral character reflect both individualizing foundations such as kindness and fairness as well as binding

foundations such as loyalty and purity [29, 30], one possibility is that heroic characters exhibit greater levels of all moral foundations compared to villainous characters. At the same time, previous findings suggest that individualizing foundations are the primary and prototypical way to conceptualize morality [19, 29–32], and are endorsed strongly by both conservative and liberal cultural groups. Accordingly, we predicted that heroes would exhibit greater levels of individualizing foundations (but not binding foundations) compared to villains.

## Method

**Movie selection.** Using decade-specific lists of top-U.S.-grossing movies, as published by IMDb.com, we selected the three top-grossing American children's movies from each five-year period beginning in 1948 (i.e., three movies from 1948–1952, 1953–1958, . . ., 2013–2017), for a total of 183 characters from 42 movies (S1 Appendix). Movies were considered "children's" movies if they 1) were rated G or PG on IMBD.com, and 2) were rated as appropriate for children 10 years old or younger on commonsensemedia.org. We used gross revenue as our selection criteria in order to identify movies with the most widespread influence.

**Coding of moral foundations characteristics.** Our coding system was adapted from the Moral Foundations Questionnaire (MFQ: [15]) to represent the extent to which each central character valued each of the five moral foundations (S2 Appendix). For each moral foundation, characters were scored on a 5-point scale. On some occasions, films presented insufficient information to illuminate a character's score on a given foundation; in such cases, data were left as missing. Scores of 5 indicated a character clearly and consistently valued the given foundation (e.g., loyalty); similarly, scores of 1 indicated a character clearly and consistently valued the foundation's inverse (e.g., betrayal). For example, to receive a care score of 5, a character met one or more of the following criteria: expresses empathy and compassion, nurtures and protects vulnerable characters, despises cruelty and exploitation, or is sensitive to emotions of others. Scores of 4 or 2 indicated a clear, though not perfectly consistent, tendency to value a foundation or its inverse. For example, if a character expressed care for all characters but one, that character received a care score of 4. Scores of 3 indicated a character's behavior was inconsistent with both the given foundation and its inverse. For example, a character who was compassionate in several scenes but cruel in a roughly equivalent number of scenes received a care score of 3.

**Characteristics coding scheme.** We also coded central characters for the following characteristics: hero/villain status, gender, race, attractiveness, weight classification (overweight vs. normal weight vs. underweight), use of violence, outcome (undesirable vs. average vs. desirable story outcome), courage, trustworthiness, accent, age, disability status, romantic involvement, socioeconomic status (SES), authority figure status, and motivation (see S3 Appendix for all definitions). For example, "desirable outcome" was defined as "happy ending; the character achieves his/her goals, or experiences happiness and pleasant events at the end of the movie."

Scores were based on the film's intended portrayal of characters rather than subjective coder opinions. For example, if a coder thought a particular character was unattractive, the coder would only rate that character as unattractive if the film clearly intended the character to be perceived as such. The default score for "attractiveness" and "weight" were average (2), and alternative scores were given only if the film clearly intended for these characteristics to depart from the norm. As with the coding of moral foundations characteristics, if a film presented insufficient information regarding a particular characteristic, we left it missing.

**Inter-rater reliability.** Before proceeding to code the entire data set (183 characters from 42 movies), two coders (both White females) independently scored 41 characters from seven randomly selected movies. Kappa exceeded .70 for the following characteristics: hero/villain

status ($k$ = .92), character outcome ($k$ = .81), gender ($k$ = 1.00), attractiveness ($k$ = .79), use of violence ($k$ = .71), accent ($k$ = .74), SES ($k$ = .75), romantic involvement ($k$ = .89), authority figure status ($k$ = .75), weight classification ($k$ = .85), disability status ($k$ = 1.00), age ($k$ = .74), and race ($k$ = 1.00). Adequate inter-rater reliability was not achieved on three characteristics (motivation, courage, and trustworthiness; all $k$ < .50), so those variables were dropped from further analysis.

Inter-rater reliability for the moral foundation characteristics was assessed in two steps. We first examined consistency between coders regarding whether a character should receive a score for a given moral foundation—that is, whether they agreed the film provided sufficient information to score the character on that dimension. Collapsing across the five foundations, Kappa for this decision was .84. We next computed an intraclass correlation coefficient (ICC) to determine agreement between coders in instances where both coders gave a numeric value. Cases in which one or both coders decided that assigning a score was unwarranted (33.2% of total cases) were excluded from this analysis. Collapsing across the five foundations, the ICC was .87, revealing strong consistency between coders. Having established agreement between independent coders, the primary coder proceeded to code the remainder of the movies, and data from that coder were used in analyses.

## Results

Data, materials, and code are publicly available via the Open Science Framework: http://doi.org/10.17605/OSF.IO/2TGM8.

**Comparing heroes and villains.** Twenty-seven characters who were neither heroes nor villains were excluded from analysis, as were five characters who began as villains and became heroes over the course of the movie. This left one hundred eighty-three characters (138 heroes, 45 villains) who fit inclusion criteria for analysis. We first examined aggregated individualizing (average of care and fairness) and binding (average of authority, loyalty, and sanctity) scores for heroes and villains (Fig 1). Two separate independent-samples $t$-tests revealed that heroes valued individualizing foundations more ($M$ = 4.27, $SD$ = 0.95) than villains ($M$ = 1.52, $SD$ = 0.77), $t(175)$ = 17.56, $p$ < .001, $d$ = 3.03 [95% CI: 2.57, 3.49], while villains valued binding foundations more ($M$ = 3.94, $SD$ = 1.19) than heroes ($M$ = 3.37, $SD$ = 1.11), $t(175)$ = 2.90, $p$ = .004, $d$ = 0.50 [95% CI: 0.16, 0.85]. Although we did not have particular hypotheses about within-character differences, we examined these patterns in two separate dependent-samples $t$-tests, finding that heroes valued individualizing foundations more than binding foundations, $t(127)$ = 7.55, $p$ < .001, $d$ = 0.67, whereas villains valued individualizing foundations less than binding foundations, $t(43)$ = -11.45, $p$ < .001, $d$ = 1.72.

We next examined each moral foundation individually (Fig 2). Heroes valued care more ($M$ = 4.25, $SD$ = 1.06) than villains ($M$ = 1.43, $SD$ = .73), $t(168)$ = 16.36, $p$ < .001, $d$ = 2.86 [95% CI: 2.41, 3.32]. Heroes likewise valued fairness more ($M$ = 4.24, $SD$ = 1.15) than villains ($M$ = 1.55, $SD$ = 1.02), $t(137)$ = 13.07, $p$ < .001, $d$ = 2.41 [95% CI: 1.95, 2.87]. Conversely, villains valued authority more ($M$ = 4.21, $SD$ = 1.09) than heroes ($M$ = 2.70, $SD$ = 1.20), $t(144)$ = 7.07, $p$ < .001, $d$ = 1.29 [95% CI: 0.91, 1.68]. Villains likewise valued sanctity more ($M$ = 3.29, $SD$ = 1.73) than heroes ($M$ = 2.00, $SD$ = 1.41), $t(28)$ = 2.24, $p$ = .033, $d$ = 0.82 [95% CI: 0.07, 1.57]. Valuation of loyalty did not differ significantly between heroes ($M$ = 4.05, $SD$ = 1.23) and villains ($M$ = 3.63, $SD$ = 1.52), $t(130)$ = 1.50, $p$ = .136, $d$ = 0.32 [95% CI: -0.10, 0.75].

**Additional characteristics as predictors of individualizing and binding scores.** We conducted exploratory analyses to determine whether individualizing and binding scores differed as a function of the additional characteristics that we coded. We excluded characteristics that had poor inter-rater reliability ($k$ < .70), had insufficient variability (e.g., 96% of characters

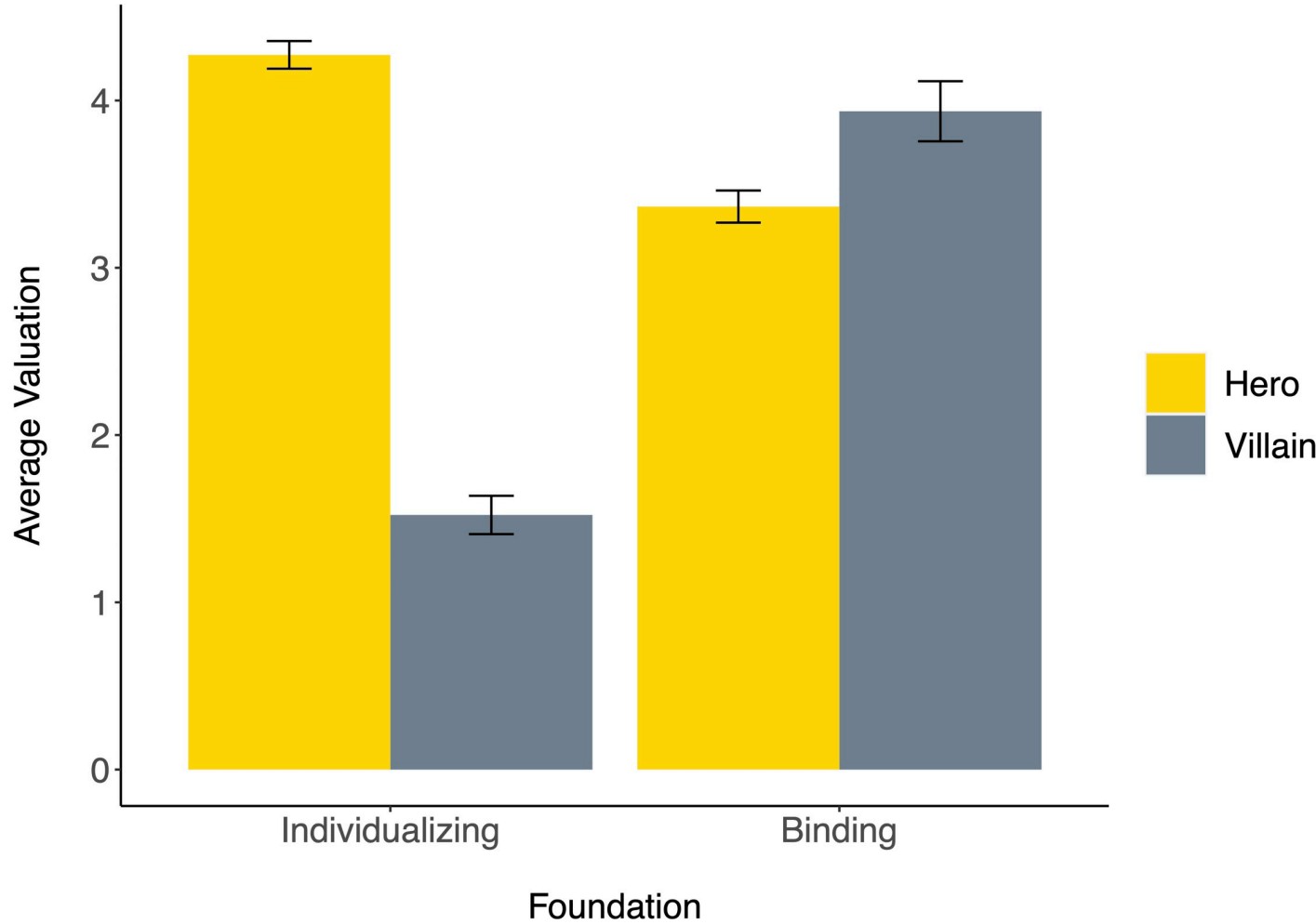

**Fig 1. Average individualizing and binding valuation among heroes and villains.** Error bars represent ± 1 SE.

had no disability), or did not have two categories that could be meaningfully compared (e.g., age had too many distinct categories/levels). This resulted in a set of 10 characteristics: character outcome, gender, attractiveness, use of violence, accent, SES, romantic involvement, authority figure status, weight, and race. For each characteristic, we conducted two $t$-tests (one for individualizing scores and one for binding scores) to determine whether these scores differed across the two applicable levels of the characteristic. In some cases, the characteristics had only two possible levels in our coding scheme (e.g., the character is vs. is not an authority figure); in some cases, we compared the two levels that represented opposing ends of a continuum (e.g., high vs. low SES, excluding average SES); and in the case of race, we compared the only two levels that were substantially represented (White vs. Non-human). To correct for multiple comparisons, we then applied the Benjamini and Hochberg [33] correction—which is less conservative than the Bonferroni correction—to the set of 10 individualizing $p$-values and the set of 10 binding $p$-values.

This approach revealed that eight of the 10 characteristics significantly predicted individualizing scores. Specifically, characters were portrayed as valuing individualizing foundations more strongly when they: experienced desirable rather than undesirable outcomes at the movie's end, $d = 3.00$; were depicted as attractive rather than unattractive, $d = 1.70$; successfully

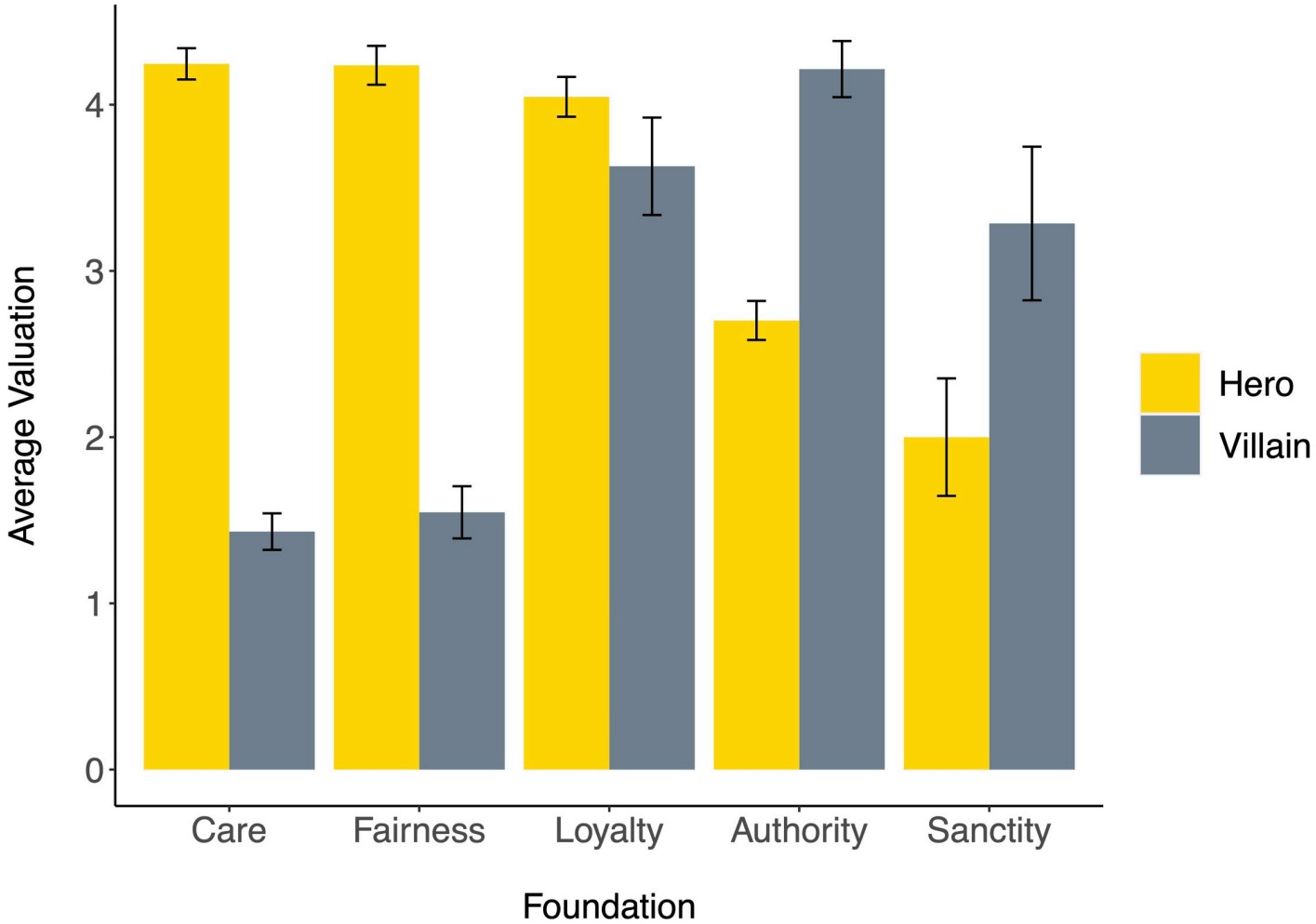

**Fig 2. Average valuation of five moral foundations among heroes and villains.** Error bars represent ± 1 SE.

rather than unsuccessfully had a romantic relationship, $d = 1.64$; were depicted as low rather than high SES, $d = 0.86$; were nonviolent rather than violent, $d = 0.57$; had an American rather than non-American accent, $d = 0.48$; were not depicted as an authority figure rather than as an authority figure, $d = 0.47$; and were female rather than male $d = 0.43$ (all corrected $p$s < .05). Individualizing scores did not differ significantly as a function of weight (overweight/obese vs. average weight) or race (White vs. non-human).

Four characteristics predicted greater binding scores: when characters unsuccessfully rather than successfully had a romantic relationship, $d = 0.99$; were depicted as an authority figure rather than not as an authority figure, $d = 0.53$; experienced undesirable rather than desirable outcomes at the movie's end, $d = 0.51$; and were nonviolent rather than violent, $d = 0.38$ (all corrected $p$s < .05). Binding scores did not significantly differ as a function of the other six characteristics (attractiveness, SES, accent, gender, weight, or race).

## Discussion

Study 1 examined the moral profiles of characters from popular American children's movies. Heroes valued individualizing foundations more than villains, and also more than they valued

binding foundations themselves. Villains valued binding foundations more than heroes, and more than they valued individualizing foundations themselves. The sole exception to these patterns was in valuation of loyalty, which did not differ significantly between heroic and villainous characters. These findings reveal that the moral messages conveyed in children's movies depict individualizing concerns of care and fairness positively, as moral virtues, whereas they largely depict binding concerns negatively.

What preferences do people themselves have regarding the moral messages presented in children's movie stories? We examined this question in Study 2 by probing people's evaluations of different moral messages—that promoted either individualizing concerns or binding concerns—in response to moral dilemmas faced by characters from children's movies. We predicted that people's preferences would depend on their political identity, and in particular that stronger political conservatism would be associated with (a) less endorsement of individualizing-promoting messages and (b) more endorsement of binding-promoting messages. Further, we predicted that correlation (a) above would be smaller in magnitude than correlation (b), reflecting relatively more agreement across the political spectrum in the importance of individualizing-promoting messages.

## Study 2

Study 2 examined whether liberals and conservatives prefer different moral content within children's movie stories. Participants self-reported their political identity and read short passages describing a moral dilemma faced by a character from a popular children's movie. They considered two hypothetical resolutions to the dilemma: one promoting individualizing foundations but violating binding foundations and the other promoting binding foundations but violating individualizing foundations. For each hypothetical resolution, participants indicated how well the resolution conveyed a morally virtuous message.

### Method

**Participants.**  We recruited participants through internet-based advertising and word-of-mouth over a 16-day period, yielding a sample of $N = 110$ (50 males, 57 females, three non-binary). Participants completed the online study voluntarily and received no compensation. The sample leaned slightly liberal ($M = 3.30$ on a 1 "strongly liberal" to 7 "strongly conservative" scale) and was primarily White ($n = 88$). No participants were excluded from analysis. All participants provided informed consent to participate and the protocol was approved by the Macalester College Institutional Review Board.

**Procedure.**  Participants read four short passages, each a scene from a children's movie where a character faced a moral dilemma. After each passage, participants read two hypothetical resolutions and rated each concerning the extent it modeled morally virtuous behavior. Participants then indicated whether they had previously seen the movie from which passages were crafted, completed the MFQ [15], and provided basic demographic information (gender, race/ethnicity, political identity). Political identity was assessed on a 1 (strongly liberal) to 7 (strongly conservative) scale, with the midpoint 4 representing "moderate".

### Materials

**Moral dilemma excerpts and resolutions.**  Each moral dilemma was followed by two hypothetical resolutions. In one resolution, the character's actions promoted individualizing foundations but violated binding foundations; the other promoted binding foundations but violated individualizing foundations. Participants rated each resolution for the extent it modeled morally virtuous behavior for children on a scale from 1 (extremely poor way to model

moral virtues) to 7 (excellent way to model moral virtues). For each participant, we excluded their ratings of any movie that they indicated that had previously seen.

For example, one passage from *Old Yeller* describes a boy deciding whether to shoot a deer to feed his family. In one hypothetical resolution, the boy chooses not to shoot the deer, thereby promoting individualizing foundations by caring for the deer's life and violating binding foundations by showing disloyalty to his family. In the other hypothetical resolution, the boy shoots the deer, promoting binding foundations but violating individualizing foundations. See S4 Appendix for all passages and resolutions.

We recruited a small pre-testing sample (N = 9) to ensure that participants interpreted the resolutions as categorized. Participants read each moral dilemma followed by each resolution (one individualizing-promoting and one binding-promoting). For each resolution, they indicated on a 1 to 6 scale (strongly disagree to strongly agree) the extent to which it conveyed the message that it was morally virtuous to (a) "protect individuals from harm and unfair treatment" (i.e., promote individualizing foundations), and (b) "be loyal to one's group and respect authority" (i.e., promote binding foundations). As expected, participants perceived that the individualizing-promoting resolutions promoted individualizing foundations ($M = 4.36$) more than binding foundations ($M = 1.42$), $t(8) = 33.32$, $p < .001$, and they perceived that the binding-promoting resolutions promoted binding foundations ($M = 5.44$) more than individualizing foundations ($M = 3.03$), $t(8) = 6.90$, $p < .001$.

**Moral foundations questionnaire.**   The MFQ [15] contains 30 items divided into two sections to assess endorsement of the five moral foundations. One section asks about the *relevance* of different moral considerations to participants' view of morality; the other asks participants to indicate their agreement with various moral *judgment* statements. An example care-relevance item reads: *Whether or not someone suffered emotionally*. An example authority-judgment item reads: *Respect for authority is something all children need to learn.* All items are answered on a 6-point scale (*not at all relevant* to *extremely relevant* for relevance items; *strongly disagree* to *strongly agree* for judgment items), with greater numbers indicating greater endorsement of the given moral foundation. Ratings were averaged across the six items for each foundation, yielding overall scores for each moral foundation.

## Results

**Moral foundations.**   We first computed individualizing-MFQ scores (Cronbach's $\alpha = .77$) for each participant by averaging harm and fairness aggregate ratings, and average binding-MFQ scores ($\alpha = .89$) by averaging loyalty, authority, and sanctity aggregate ratings. Individualizing and binding aggregate MFQ scores were weakly and non-significantly negatively correlated, $r(108) = -.12$, $p = .20$.

We examined political identity as a predictor of individualizing- and binding-MFQ scores in two separate linear regression models (Fig 3). Consistent with previous findings, stronger conservatism predicted lower individualizing-MFQ scores, $\beta = -.27$, $t(108) = -2.91$, $p = .004$, $R^2 = .07$, and higher binding-MFQ scores, $\beta = .58$, $t(108) = 7.42$, $p < .001$, $R^2 = .33$. A comparison of the magnitude of these correlations—ignoring their sign—using the R-package *cocor* [34] revealed that the magnitude differed significantly: political identity was more strongly associated with binding- than individualizing-MFQ scores, 95% CI of difference = [.10, .52] [35]. This comparison is conceptually similar to an interaction between political identity and moral foundation category; in the Supporting information, we report the results of this interaction analysis, which leads to the same conclusion.

**Story resolution scores.**   We computed overall individualizing-resolution ($\alpha = .84$) and binding-resolution scores ($\alpha = .79$) for participants by averaging ratings of the corresponding

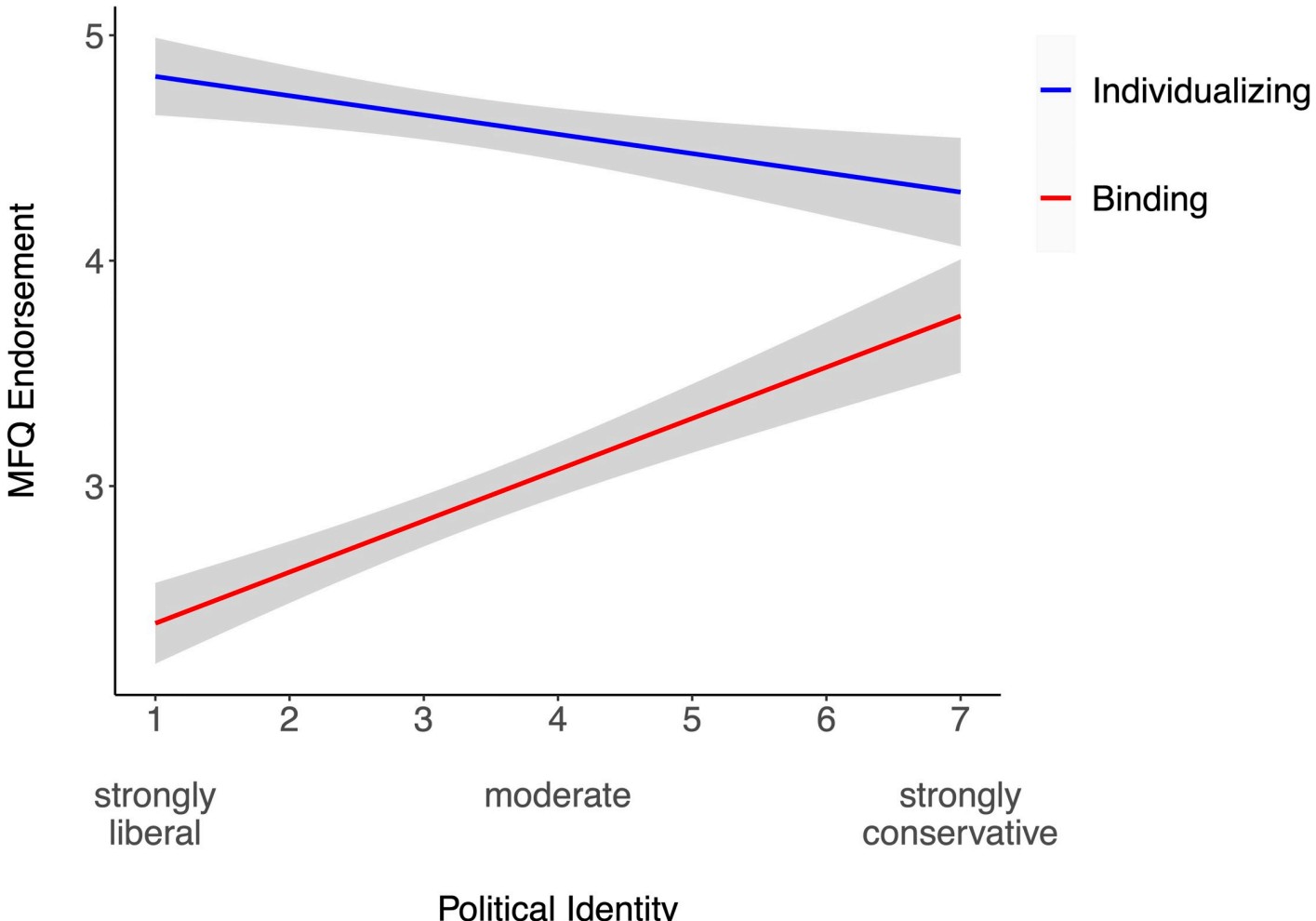

**Fig 3. Association between political identity and endorsement of MFQ individualizing and binding moral foundations.** The plot depicts the 95% CI around each regression line.

resolution across the four passages. Ratings were excluded if participants indicated they had seen the given film previously. Across the sample, participants expressed moderate endorsement of both individualizing ($M = 4.53$, $SD = 1.48$) and binding ($M = 4.05$, $SD = 1.44$) story resolutions.

Greater endorsement of binding resolutions was associated with greater binding-MFQ scores, $r = .60$, $t(106) = 7.74$, $p < .001$ but not with individualizing-MFQ scores, $r = -.04$. Greater endorsement of individualizing resolutions was associated with lower binding-MFQ scores, $r = -.28$, $t(106) = 3.05$, $p = .002$; surprisingly, endorsement of individualizing resolutions was not associated with individualizing-MFQ scores, $r = .03$.

We examined political identity as a predictor of individualizing- and binding-resolution scores in separate linear regression models (Fig 4). Stronger conservatism predicted less endorsement of individualizing resolutions, $\beta = -.31$, $t(106) = -3.37$, $p = .001$, $R^2 = .09$ and more endorsement of binding resolutions, $\beta = .57$, $t(106) = 7.14$, $p < .001$, $R^2 = .32$. Comparing the magnitude of these correlations revealed that political identity was more strongly associated with endorsement of binding than individualizing resolutions, 95% CI of difference = [.11, .41].

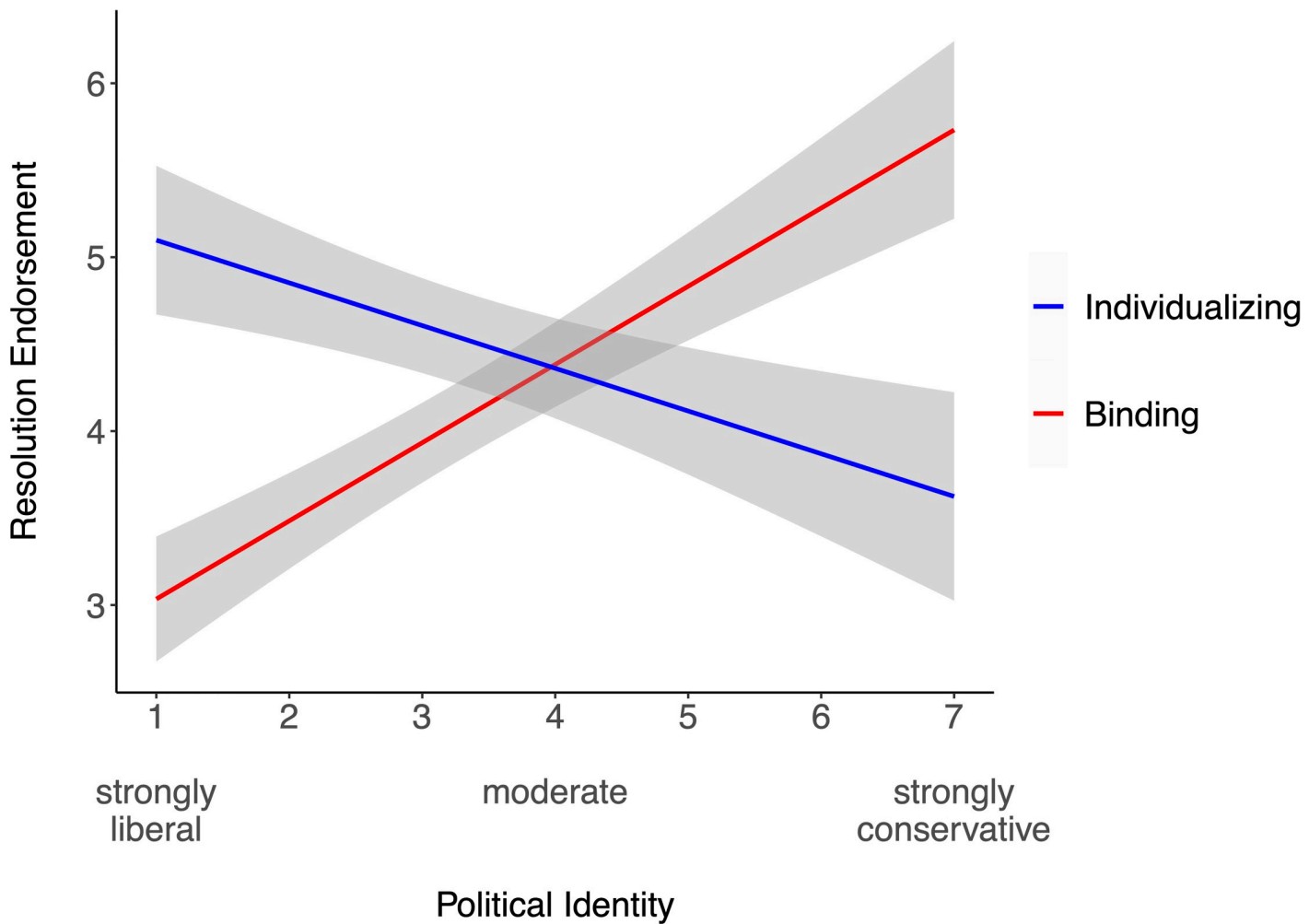

**Fig 4. Association between political identity and endorsement of individualizing and binding moral resolutions.** The plot depicts the 95% CI around each regression line.

Two final regression models predicted resolution endorsement with political identity and both individualizing- and binding-MFQ aggregates as predictors. Stronger conservatism negatively predicted endorsement of individualizing resolutions, $\beta$ = -.23, $t(104)$ = -1.99, $p$ = .048 and positively predicted binding resolutions, $\beta$ = .36, $t(104)$ = 3.93, $p < .001$ while controlling for MFQ scores.

## Discussion

Study 2 examined political identity as a predictor of moral dilemma resolution preference. Replicating well-established findings [14, 15], stronger political conservatism was associated with weaker individualizing-MFQ endorsement and stronger binding-MFQ endorsement. Further, we extended previous findings to endorsement of moral messages in children's movies. When presented with moral dilemmas from unfamiliar children's movies, individuals with stronger self-reported political conservatism evaluated binding-promoting moral messages more favorably and individualizing-promoting messages less favorably.

Our findings regarding moral resolution preferences converge with the broader MFT literature while also serving to replicate conceptually the key predictions of MFT, in particular by

revealing that political identity is systematically associated with endorsement of moral messages. Notably, the link between political identity and resolution endorsement was not reducible to MFT preferences, as it remained strong even after adding MFT scores as simultaneous predictors in multiple regression models. Thus, political identity predicted differential valuing of individualizing- vs. binding-promoting moral lessons in a way that was not merely accounted for by people's general preference for individualizing vs. binding concerns.

Study 2 further revealed that political identity was more strongly associated with binding than individualizing concerns (for both MFT preferences and resolution endorsement), indicating relatively more agreement across the political spectrum in the importance of individualizing foundations. This confirms previous research demonstrating that liberals and conservatives both value individualizing foundations at an automatic, intuitive level [19], and that individualizing concerns—and concerns about harm more specifically—might be the single unifying lens through which all people interpret moral violations [20, 31, 32].

## General discussion

The present two studies drew on key findings of Moral Foundations Theory to examine how movies themselves depict the moral characteristics of heroes and villains and how people prefer different moral lessons in children's movies depending on their political identity. To determine whether Americans' generalized support of individualizing foundations is reflected in the moral messaging of children's movies, Study 1 examined the moral characteristics of heroic and villainous characters from widely-viewed American children's movies. It revealed that heroes valued the individualizing foundation more than villains, whereas villains tended to value the binding foundations more than heroes. Study 2 revealed that stronger political conservatism predicted more preference for binding-promoting moral resolutions in children's movie scripts and less preference for individualizing-promoting moral resolutions. Additionally, the results showed that the relationship between political identity and moral preferences was weaker for individualizing resolutions than for binding resolutions. Thus, there was more shared agreement across political identity in the importance of individualizing concerns. The findings of Studies 1 and 2 therefore dovetail, revealing that the portrayal of heroes' moral foundations in children's movies mirrors the moral foundations of people in general.

The present results are consistent with previous research on MFT, which shows that political identity predicts moral preferences [36], and specifically that stronger political conservatism predicts more concern for binding foundations and less concern for individualizing foundations [14, 15]. Our findings also correspond with previous research suggesting that despite political differences, there is stronger agreement across the political spectrum in the importance of individualizing over binding foundations [19, 32]. Furthermore, the fact that MFT accounts for moral depictions and preferences in children's media adds to a growing body of research that MFT predicts moral judgments in naturalistic contexts, including people's judgments about social justice protests [37], environmental protection [38], and decisions about whether to engage in whistleblowing [39].

### Depictions of morality among heroes and villains

Study 1 offers insight into the ways children's movies depict the moral characteristics of heroes and villains and therefore contribute to children's moral education. Movies depict values of compassion and fairness as virtuous by imbuing heroes (but not villains) with these qualities, and by portraying heroes as fighting for the rights and well-being of others, even when doing so entails physical or social risks. Empathy and compassion play pivotal roles in moral development and prosocial behavior [40], and extreme lack of empathy is a hallmark of

psychopathy [41, 42]. Our findings suggest that depictions in children's movies mirror these patterns, conveying that good characters aim to minimize suffering and injustice.

While embodying these individualizing moral virtues, heroes often simultaneously disobey authority figures, typically in the name of protecting vulnerable individuals (e.g., Mike and James from "Monsters, Inc." defy the rules of their boss to save Boo, a lost toddler). Conversely, villainous characters are generally submissive even to tyrannical authority figures (e.g., Gaston's henchman LeFou from "Beauty and the Beast" persistently ingratiates himself to Gaston despite Gaston's callousness). The fact that respect for authority is not typically presented as a virtue may reflect American individualistic cultural norms, wherein the autonomy of individuals eclipses the value of individuals' interrelationships with others [43]. In individualistic cultures, it is admirable to "speak truth to power" and to fight for one's beliefs, especially when those beliefs contradict prevailing norms or dictates of authority figures. It seems likely, therefore, that in less individualistic cultures, heroic characters might be depicted with a stronger valuation of authority.

It is possible—and indeed likely—that the differential endorsement of moral foundations among heroes and villains is the very evidence that people use to categorize these characters as such. (We thank an anonymous reviewer for raising this possibility.) That is, endorsing the binding foundations at the expense of the individualizing foundations might be the very pattern that tells the audience (at least an American one) that a character is a villain. Future research could assess this possibility more directly by determining at what point in a narrative people categorize the characters as heroes/villains, and how this relates to the characters' valuation of moral foundations up until that point.

Contrary to our hypotheses, heroes and villains were depicted with equally high valuation of loyalty. However, a closer inspection suggests that loyalty might manifest in different ways among children's movie heroes versus villains. Heroes are generally loyal to immediate friends and family. For example, in "Finding Nemo," hero Dory endangers herself to help her good friend Marlin. A distinct manifestation of loyalty, however, is to show loyalty toward one's larger social group. Villains tend to show this type of loyalty, for example when "Peter Pan's" Mr. Smee remains a loyal member of Captain Hook's crew despite being mistreated by them. Thus, loyalty might buttress individualizing concerns among heroes, but buttress binding concerns among villains. This nuance in loyalty's manifestations concords with the tendency of children's movies to portray individualizing foundations favorably and binding foundations unfavorably.

## Media transmission of moral foundations

Although our overall findings showed that individualizing foundations are represented more positively than binding foundations in children's movies, there undoubtedly is variation in this pattern. Some movies portray binding foundations favorably, while others emphasize the importance of individualizing foundations, and thus there is room for people to prefer some movies over others based on moral content. Indeed, the results of Study 2, consistent with previous MFT research [14], revealed that whereas liberals preferred moral messages that promoted individualizing foundations, conservatives preferred moral messages that promoted binding foundations.

Taken together, the present research suggests differential media exposure might be one pathway by which children develop and adopt political and moral views that model those of their parents. Since stories are influential to moral development [24, 44] and moral values are tightly linked to political identity [14], viewing morally-charged movies might shape children's development of moral and political preferences. Parents likely select and suggest the media

their young children are exposed to, perhaps preferring media that conforms to their own values. More liberal parents might select media that promotes individualizing moral messages ("Robin Hood," for example), while conservative parents might select media that emphasizes loyalty and respect for authority as virtuous ("Old Yeller," for example). Indeed, the Moral Intuition and Media Entertainment model (MIME; [45]) argues for this sort of reciprocal relationship between media content and moral intuitions, whereby adults selectively consume media that aligns with their moral intuitions, and these intuitions are themselves influenced by the moral content of the media they consume. Future research should continue to investigate these pathways, particularly as they relate to children's media consumption and moral development.

## Future directions

We focused on moral depictions specifically in children's movies because childhood is a critical period for moral education and development, but movies targeted for adult audiences often likewise present moral depictions and messages. How might moral depictions in movies for children differ from those in movies for adults? Given the widespread importance of individualizing foundations, we suspect that just as in children's movies, heroes in adults' movies would show greater concern for harm and fairness compared to villains. However, there might be a less striking difference in the moral profiles of heroes vs. villains in adults' movies as compared to children's movies. Children's movies sometimes eliminate moral nuance from their characters, presenting heroic characters that are unambiguously moral [46]. Heroes in adults' movies often must navigate more complex moral decisions, committing particular moral infractions (lying; violence) to achieve moral ends. Consequently, heroes in adults' movies might be perceived as less obviously or unambiguously good, and their moral profiles might show greater overlap with those of villainous characters.

Future research could also examine more fine-grained preferences for particular moral messages in children's media. In Study 2, we investigated moral resolutions that promoted individualizing vs. binding foundations, but we did not isolate specific foundations within these categories. Our binding-promoting resolutions arguably promoted both loyalty and obedience to authority simultaneously (e.g., Travis' decision in *Old Yeller* to follow his mother's order and shoot the deer to feed their family, thereby demonstrating both loyalty to family and obedience to authority). Given Study 1's findings that heroes showed strong valuation of loyalty, it is possible that both liberals and conservatives would show broad agreement in the importance of moral messages promoting loyalty. However, political identity likely predicts endorsement of different *types* of loyalty. Since conservatives prefer more tight social connections than liberals [47], conservatives are likely to prefer messages promoting loyalty to smaller groups such as one's family, whereas liberals are likely to prefer messages promoting loyalty to broader or more diffuse social groups.

Finally, we focused specifically on morality in American movies, but there is bound to be substantial variability in moral depictions across cultures. Collectivistic cultures typically value interdependence, whereas individualistic cultures typically value independence [43]. Consequently, heroic characters from collectivistic cultures are likely to show greater endorsement of the binding foundations, and less endorsement of the individualizing foundations, as compared to our findings from American movies in Study 1. At the same time, the correlation between political identity and MFT endorsement is robust across both individualistic and collectivistic cultures [15]. Therefore, we suspect that even in collectivistic cultures, the correlational patterns from Study 2—showing that stronger conservatism predicted more endorsement of binding messages and less endorsement of individualizing messages—would continue to hold.

## Conclusion

Two studies investigated Moral Foundations Theory as related to people's preference for different moral messages in children's movies and the depictions of moral characteristics in such movies. Study 1 found that heroic characters are depicted as valuing the individualizing concerns of care and fairness far more than villains, reflecting the broad preference for those moral messages, at least in an American context. Study 2 found that although political conservatives preferred individualizing moral lessons less than liberals and preferred binding moral lessons more than liberals, the discrepancy was weaker for individualizing preferences, indicating more agreement across the political spectrum in the importance of individualizing moral foundations. Despite political polarization in the U.S, children's movies may be a place of commonality, where the moral concerns of care and fairness are celebrated by both liberals and conservatives.

## Supporting information

**S1 Appendix. Movies included in Study 1.**
(DOCX)

**S2 Appendix. Moral foundations coding scheme [48, 49].**
(DOCX)

**S3 Appendix. Additional characteristics coding scheme [25].**
(DOCX)

**S4 Appendix. Moral dilemma and resolution task.**
(DOCX)

**S5 Appendix. Analyses reported in Study 1.**
(DOCX)

**S6 Appendix. Analyses reported in Study 2.**
(DOCX)

**S7 Appendix. Supplementary analyses for Study 2.**
(DOCX)

## Author Contributions

**Conceptualization:** Rachel Gehman, Steve Guglielmo, David C. Schwebel.

**Formal analysis:** Steve Guglielmo.

**Methodology:** Rachel Gehman, Steve Guglielmo, David C. Schwebel.

**Writing – original draft:** Rachel Gehman, Steve Guglielmo.

**Writing – review & editing:** Rachel Gehman, Steve Guglielmo, David C. Schwebel.

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
