## [Decision Letter · Decision Letter 0]

25 Sep 2020

PONE-D-20-26290

Moral Foundations Theory, Political Ideology, and the Depiction of Moral Values in Children’s Movies

PLOS ONE

Dear Dr. Guglielmo,

Thank you for submitting your manuscript to PLOS ONE. After careful consideration, we feel that it has merit but does not fully meet PLOS ONE’s publication criteria as it currently stands. Therefore, we invite you to submit a revised version of the manuscript that addresses the points raised during the review process.

We look forward to receiving your revised manuscript.

Kind regards,

Valerio Capraro

Academic Editor

PLOS ONE

Journal Requirements:

2. Please change "female” or "male" to "woman” or "man" as appropriate, when used as a noun.

Additional Editor Comments (if provided):

I have now collected two reviews from two experts in the field. The reviews are somehow split, as one recommends major revision and one recommends rejection. However, reading through the negative review, I have convinced myself that the issues raised by the reviewer are mainly structural and can perhaps be addressed in a major revision. For this reason, I would like to invite you to revise your work for Plos One. I would like to stress the fact, however, that this revision is probably riskier than the average decision, as, if the disagreement between the reviewers persists, I might invite a third reviewer.

Reviewers' comments:

Reviewer's Responses to Questions

**Comments to the Author**

1. Is the manuscript technically sound, and do the data support the conclusions?

Reviewer #1: Partly

Reviewer #2: Partly

2. Has the statistical analysis been performed appropriately and rigorously? 

Reviewer #1: No

Reviewer #2: Yes

3. Have the authors made all data underlying the findings in their manuscript fully available?

Reviewer #1: Yes

Reviewer #2: Yes

4. Is the manuscript presented in an intelligible fashion and written in standard English?

Reviewer #1: Yes

Reviewer #2: No

5. Review Comments to the Author

Reviewer #1: The current paper investigates the moral code in children’s movies: one hand it looks what moral codes are conveyed to children’s in the form of movie, on the other it looks at adults’ preferences for the moral codes being transmitted: supporting individualizing versus binding moral foundations.

My overall impression is positive: the topic is interesting and the writing is engaging but my main concerns are related to the generalizability of their findings and the samples under study, formulation of hypotheses in both studies, and several minor issues.

Generalizability and sampling

1. In Study 1, the authors first choose the most popular movies per decade and then sample only one per decade to analyze (7 movies in total have been analyzed). I am not an expert in the content analysis of this kind but a rather small sample of movies being analyzed made me wonder of how “representative” it might be for children’s movies in US/North America as such. Especially given the fact that the movies were selected from such a broad period of time. Is it a standard procedure to select only a few movies for the analysis?

Although the analysis of changes in the characters and the moral codes they have been conveying could be an interesting follow up, I was wondering if it would not serve the purpose of the study better if the authors focus on the analysis of only the recent movies?

2. I have also concerns related to the sample in Study 2:

2a. In my opinions, important details are missing with regard to who has been invited to take part in Study 2 and the final sample (what is the age, education, and nationality of the participants?).

2b. How the sample size was determined? E.g., what was the stopping rule for the data collection?

These two issues make me hesitant about whether we can draw strong conclusions from this research.

3. Although the study is ran in the US context, this issue has not been raised much (e.g., as mentioned earlier, the nationality of the sample is not stated in Study 2). Although the authors suggest that the findings might be different in the collectivistic cultures, in my opinion this discussion was framed in a way that the “default” option of how things work is presented in the current paper and other cultures or contexts are some sort of exceptions (see for example, Cheon, Melani, & Hong, 2020). I find this problematic and would like to encourage authors to think about what are the boundaries to the generalizability of their findings.

Cheon, B. K., Melani, I., & Hong, Y. Y. (2020). How USA-Centric Is Psychology? An Archival Study of Implicit Assumptions of Generalizability of Findings to Human Nature Based on Origins of Study Samples. Social Psychological and Personality Science, 11(7), 928-937.

Hypotheses

4. The hypotheses to Study 1 are not clear to me: on p. 7 and also on p. 8, they refer to the differences between heroes and villains (e.g., p.7: “heroes would value individualizing foundations more strongly than villains, whereas the opposite would be true for binding foundation”), whereas later in the Results section, the authors present between-characters differences as well as within-characters differences for the two kinds of moral foundations. Please clarify what kind of differences (between- or within-protagonists) you were expecting and why.

5a. Furthermore, when the foundations are analyzed separately, then authors focus only on the differences between heroes and villains but not differences within the two types of characters.

5b. Relatedly, the analysis for the Study 1 could be, in my opinion, more elegant if instead of running three t-tests (p. 11), the authors run on ANOVA with an interaction.

5. Similarly in Study 2: it is not clear whether the authors expected between- or within-political groups differences. The hypothesis suggests, it is between-groups differences that are expected (p. 14: “We predicted that people’s preferences would depend on their political identity, and in particular that political liberals would prefer individualizing-promoting messages whereas political conservatives would prefer binding-promoting messages.”) but in the analysis, the authors compares the two foundations with each other.

Additionally, why do authors run two separate regressions for the individualizing and binding MFQ scores (p. 17) and resolution preferences (p. 18)? Running two separate analysis and then comparing the strength of the correlation seem to be a non-standard procedure. Why did authors choose this approach? Instead, I would expect a regression analysis with an interaction (e.g., using mixed models because of the repeated measures).

It is also not clear to me how the comparison between the correlations worked. For example, on p. 18, the authors claim the magnitude of the correlations was larger for the binding than individualizing foundations. Does this analysis provides information about differences in strength of the absolute values of the correlations or does it test whether the two correlations significantly differ from each other? If it is the latter, the conclusions about the importance of individualizing moral foundations across political ideology may not be justified because the correlation coefficients had different directions.

Loyalty

6. I was wondering what characteristics were predictive of loyalty in Study 1 as the differences between villains and heroes were not significant (and looking at Figure 2, there might have been also within-characters differences for this this foundation)? Analysis on pp. 13-14 is presented for the two overarching foundations only.

7. I was surprised that in Study 2 the authors analyzed the binding vs. individualizing foundations without looking into the effects of loyalty, given the above mentioned findings for loyalty.

Other comments

- Throughout the text (e.g., p. 5), the authors refer to “critical claim of MFT” or “key claims of MFT”, what are these claims?

- Please list all the dimensions that were used to describe movie characters in Study 1 to the section “Characteristics coding scheme” instead of directing the readers to the Appendix,

- Bonferroni correction used in Study 1 might be over-conservative and reduce statistical power (e.g., Narum, 2006; Perneger, 1998). Authors may look into other approaches to adjust for multiple testing (e.g., Benjamini-Hochberg procedure, 1995, or other methods),

Benjamini, Y., & Hochberg, Y. (1995). Controlling the false discovery rate: a practical and powerful approach to multiple testing. Journal of the Royal Statistical Society: Series B (Methodological), 57(1), 289-300.

Narum, S. R. (2006). Beyond Bonferroni: less conservative analyses for conservation genetics. Conservation genetics, 7(5), 783-787.

Perneger, T. V. (1998). What's wrong with Bonferroni adjustments. BMJ, 316(7139), 1236-1238.

- I think APA standards ask for 2 decimal numbers when presenting statistics (even descriptive ones),

- Also regarding APA standards, I think results for the Cohen’s d should be reported with leading 0 (as the values can go higher than 1).

- Please provide details for the statistical analysis for t-tests on p.13 as well as report full findings for the regression analysis on p. 18 (last paragraph). They could be reported in the Supplementary materials.

- I was also wondering whether the current findings could be expected in other media? For example, children’s literature or comic books? And relatedly, should be expect different effects in the movies for teenagers or adults? Why children’s movies are especially important?

Reviewer #2: Please see attached comments

6. PLOS authors have the option to publish the peer review history of their article (what does this mean?). If published, this will include your full peer review and any attached files.

Reviewer #1: No

Reviewer #2: No

---

## [Author Response · Author response to Decision Letter 0]

24 Feb 2021

All responses to reviewer and editor comments are included in the enclosed 'Response to Reviewers' document

---

## [Editor Report · Decision Letter 1]

9 Mar 2021

Moral Foundations Theory, Political Identity, and the Depiction of Morality in Children’s Movies

PONE-D-20-26290R1

Dear Dr. Guglielmo,

We’re pleased to inform you that your manuscript has been judged scientifically suitable for publication and will be formally accepted for publication once it meets all outstanding technical requirements.

Kind regards,

Valerio Capraro

Academic Editor

PLOS ONE
---

## [Editor Report · Acceptance letter]

16 Mar 2021

PONE-D-20-26290R1 

Moral Foundations Theory, Political Identity, and the Depiction of Morality in Children’s Movies 

Dear Dr. Guglielmo:

I'm pleased to inform you that your manuscript has been deemed suitable for publication in PLOS ONE. Congratulations! Your manuscript is now with our production department. 

Kind regards, 

on behalf of

Dr. Valerio Capraro 

Academic Editor

PLOS ONE